Coronary artery disease classification using ConvMixer based classifier from CT angiography images

Rajeev C.
Natarajan Karthika karthika.n@vitap.ac.in
School of Computer Science and Engineering, VIT-AP University , Amaravati, Andhra Pradesh , India
Coelho Paulo Jorge
Electronic publication date: 2025 Mar 27
Publication date: 2025
Volume: 11
Electronic Location ID: e2771
Received 2024 Nov 5; Accepted 2025 Feb 25
Copyright: © 2025 Rajeev and Natarajan
Copyright year: 2025
Copyright holder: Rajeev and Natarajan
License: This is an open access article distributed under the terms of the Creative Commons Attribution License, which permits unrestricted use, distribution, reproduction and adaptation in any medium and for any purpose provided that it is properly attributed. For attribution, the original author(s), title, publication source (PeerJ Computer Science) and either DOI or URL of the article must be cited.
License URL: https://creativecommons.org/licenses/by/4.0/

Keywords: Coronary artery disease, Convmixer, Angiography, Deep learning, Computed tomography

Funding: The authors received no funding for this work.

==============================
Coronary artery disease (CAD) has recently emerged as a predominant source of morbidity and death worldwide. Assessing the existence and severity of CAD in people is crucial for determining the optimal treatment strategy. Currently, computed tomography (CT) delivers excellent spatial resolution pictures of the heart and coronary arteries at a rapid pace. Conversely, several problems exist in the analysis of cardiac CT images for indications of CAD. Research investigations employ machine learning (ML) and deep learning (DL) techniques to achieve high accuracy and consistent performance, hence addressing existing restrictions. This research proposes convMixer with median filter and morphological operations for the classification of the coronary artery disease from computed tomography angiography images. A total of 5,959 CT angiography images were used for classification. The model achieved an accuracy of 96.30%, sensitivity of 94.39%, and specificity of 99.16% for combination of the morphological operations and convMixer, 88.92% of accuracy and 89.56% of sensitivity, and 93.10% of specificity for the combination of median filter and convMixer and 94.63% of accuracy, 95.82% of sensitivity, and 93.10% of specificity for convMixer. The findings indicate the viability of automated non-invasive identification of individuals necessitating invasive coronary angiography images and maybe future coronary artery operations. This may potentially decrease the number of people who receive invasive coronary angiography images. Lastly, post-image analysis was conducted using DL heat maps to understand the decisions made by the proposed model. The proposed integrated DL intelligent system enhances the efficiency of illness diagnosis, reduces manual involvement in diagnostic processes, supports medical professionals in diagnostic decision-making, and offers supplementary techniques for future medical diagnostic systems based on coronary angioplasty.

Introduction

Coronary artery disease (CAD) are the leading cause of disability and early mortality in the European region, accounting for about 42.5% of annual fatalities. This equates to 10,000 fatalities daily (WHO, 2024). The WHO/Europe research indicates that males in the region are about 2.5 times more susceptible to mortality from cardiovascular diseases than women. The regional disparity is evident, as the likelihood of dying prematurely (ages 30–69) from cardiovascular disease is about five-fold greater in Eastern Europe and Central Asia than in Western Europe. “Cardiovascular diseases and hypertension are predominantly preventable and manageable,” stated Dr. Hans Henri P. Kluge, WHO Regional Director for Europe. “Four million, an astonishing statistic, represents the annual fatalities attributed to cardiovascular diseases, predominantly affecting men, especially in the eastern sector of our WHO region.” These facts highlight the urgency for change. Despite understanding effective strategies, the failure to consistently apply evidence-based methods continues to result in unacceptably high rates of preventable fatalities. Implementing targeted efforts to decrease salt consumption by 25% might potentially save around 900,000 fatalities from cardiovascular diseases by 2030 (Kryuchkov, 2024). The number of victims is increasing dramatically. As a result, healthcare facilities must develop a mechanism for early detection of CAD. Recent advancements in convolutional neural networks (CNN) models allow researchers to create a predictive model for CAD (Xu et al., 2021; Nishi et al., 2021; Gülsün et al., 2016; Alizadehsani et al., 2019; Liu et al., 2021b). Nonetheless, CNN’s architecture is intricate and requires a high-performance graphics processing unit (GPU) to handle complicated pictures. Traditional methods regard analytical angiography as one of the most precise techniques for identifying cardiac anomalies. Angiography’s drawbacks are its high cost, potential side effects, and the requirement for advanced technological expertise (Banerjee, Ghose & Mandana, 2020). Conventional procedures frequently produce erroneous diagnoses and require extended time-frames due to human error. Furthermore, it is an expensive and labor-intensive approach to illness diagnosis that requires significant processing. Clinical diagnostic systems have gradually incorporated Artificial Intelligence (AI) technologies over the past three decades to improve their precision. In recent years, data-driven decision-making utilizing AI algorithms has become increasingly prevalent in the CAD sector (Zreik et al., 2018b). Automation and standardization of interpretation and inference procedures can enhance diagnostic accuracy. AI-driven technologies can expedite decision-making processes for overcoming the shortcomings of existing approaches. Visual evaluation of coronary CT angiography images is subjective and can vary from observer to observer. In contrast, intrusive methods such as invasive coronary angiography (ICA) require a lot of resources and come with their own risks. AI-driven solutions can standardize interpretations, diminish diagnostic mistakes, and accelerate decision-making, therefore enhancing patient outcomes. Still, the fact that AI models are not always applicable is a big problem. This is because many models are only trained on small datasets and might not work well with different types of people or imaging methods. AI models that cannot be interpreted are hard to use in clinical settings because doctors need clear ways to make decisions in order to trust and use these technologies effectively. Healthcare centers may acquire, assess, and analyze data from these developing technologies to enhance patient services (Wolterink et al., 2019). The raw data can profoundly influence the quality and efficacy of AI methodologies. Consequently, substantial collaboration between AI engineers and healthcare practitioners is essential to enhance diagnostic quality (Papandrianos & Papageorgiou, 2021). The novel CAD detection method relies on images. Eliminating unnecessary characteristics enables physicians and computer scientists to provide predictions more rapidly. The essential characteristics of CAD determine the efficacy of AI methodologies (Mamun & Alouani, 2020). Numerous studies employ DL to ascertain the presence of CAD (Lin et al., 2021; Liu et al., 2021a; Morris & Lopez, 2021; Rim et al., 2021; Cho et al., 2021). The main objective of this research is to develop an automated and efficient diagnostic tool that leverages the advantages of CNNs, MLP-mixer, and ViTs for the detection and classification of CAD from coronary CT angiography images.

CNN have been the standard architecture for DL techniques utilized in computer vision applications for a number of years. Transformer-based designs, like the vision transformer (ViT) architecture (Dosovitskiy, 2020), have recently shown impressive performance in several applications, often outperforming conventional convolutional networks, especially when dealing with big data sets. To use transformers for images, their representation must be modified; applying self-attention layers in transformers directly at the per-pixel level would cause computational costs to rise quadratically with the number of pixels in each image. The solution, therefore, involves dividing the image into multiple patches, linearly embedding each one, and then applying the Transformer to this collection of patches.

A basic convolutional architecture called ConvMixer was proposed to assist in classifying CAD from CT angiographic images. It is very similar to ViT and MLP-mixer (Tolstikhin et al., 2021). It uses direct patch processing, keeps the size and resolution of the representation the same across all layers, stops representation down-sampling in later layers, and can tell the difference between channel-wise and spatial mixing of data. In contrast to the ViT and Multi-Layer Perceptrons Mixer (MLP-Mixer), our solution simply uses regular convolutions to do all of these tasks. The major contributions of this work are: Used a large dataset of 5,959 CT angiography images.

Median filtering (MLFR) and morphological (Morpho) operations are used to pre-process the CT angiography images to remove the noise and sharpen the edges and enhancement in images.

ConvMixer architecture is proposed for the classification by extraction of patches similar to the ViT and MLP-mixer. This is the first time the ConvMixer architecture has been used for classifying CAD.

Done the performance evaluation with the help of the performance metrics such as accuracy, sensitivity, specificity, F-score, precision, Jaccard Index (JCI), Kappa and Matthew’s coefficient (MC).

Applied Explainable Artificial Intelligence (XAI) methods, such as Gradient-weighted Class Activation Mapping (Grad-CAM), Local Interpretable Model-agnostic Explanations (LIME) and occlusion sensitivity (OS) to the images to interpret the decision-making process of the ConvMixer model.

Literature review

In the healthcare industry, ML is rapidly becoming a transformative instrument for improving patient diagnoses. This is an analytical approach for extensive and complex programming jobs, encompassing data translation from medical records, pandemic prediction, and genetic data analysis. Numerous studies proposed various methodologies for detecting cardiac problems using ML (Overmars et al., 2022; Lei et al., 2020; Kawasaki et al., 2020). The ML methodology involves numerous stages, such as picture preprocessing, feature extraction, model training and parameter optimization, model evaluation, and ultimately, the generation of predictions using the models. The classifier’s efficacy is contingent upon the feature selection procedure. The contemporary literature (van Hamersvelt et al., 2019) has delineated numerous criteria for the assessment of the ML-based model. Healthcare practitioners are mostly concerned with the reliability and performance of the ML-based model. Furthermore, ease of use, accessibility, and computational difficulties are critical factors for implementing the CAD detection model in healthcare facilities (Al-Aref et al., 2019). DL is an emerging ML approach with significant potential for diverse categorization challenges. DL provides an effective methodology for constructing a comprehensive model that utilizes raw medical images to forecast a significant illness (Karaddi, Sharma & Bhattacharya, 2024; Karaddi & Sharma, 2023). The CNN model surpasses alternative approaches in certain picture classification tasks. CNN delineates the essential attributes and categorizes photos (Lu et al., 2022). Picture annotation is a crucial element in medical picture categorization. Elevated dataset dimensionality is a significant challenge for ML methodologies (Kolossváry et al., 2019). Assigning weights to features, minimizing duplicate data, and mitigating overfitting can enhance the algorithm’s efficacy (Mathur et al., 2020; Paul et al., 2022; Dong, Xu & Li, 2022).

Abdar et al. (2019) proposed automatic CAD detection using the N2-generation-NVSVM model. For the classification, the authors experimented with ten different ML models and used 10-fold cross validation for the parallel selection of features and training models. The authors achieved an accuracy of 93.08% for the N2-generation-NVSVM model for the classification. Saeedbakhsh et al. (2023) suggested a CAD diagnostic tool based on unsupervised learning models such as SVM, random forest, and artificial neural networks. The authors used 11,495 CT angiography images for the classification. Using SVM, they achieved an accuracy of 89.73%. Sayadi et al. (2022) classified the CAD using Z-Alizadaly image datasets, employing six ML algorithms: decision tree, DL, SVM, Xgboost, random forest, and logistic regression. To diagnose CAD, the authors used a Pearson feature selection model with eight features. They achieved the highest accuracy for SVM, 95.45%. Garavand et al. (2022) used clinical parameters and angiography images to divide CAD into groups using various ML models, including SVM, k-nearest neighbors, and multi-layer perceptron. To advise medical professionals, the authors used 303 records with 25 features. The authors achieved an accuracy of 88% and f-measure of 88% using SVM. Muthusamy & Murugesh (2024) used modified DenseNet201 for feature extraction and segmentation and ResNet152 for CT angiography image classification. For classification, Jin et al. (2022) used 505 patients with 127 and 763 CT angiography images. The authors used CNN for plaque segmentation and detection. The authors also used DL-based techniques to extract the patches from the image. The authors use decision trees and gradient boosting as classifiers to classify the illness based on the extracted features. The authors achieved 87% of the accuracy, 84.1% of sensitivity, and 95.7% of the specificity. Zreik et al. (2019) analyzed the CAD using DL techniques derived from angiography images. In 192 different arteries, the authors used 187 patients and 137 invasive fractional flow reverse measurements. The authors achieved 87% accuracy for CAD detection. Zreik et al. (2018a) proposed an automatic classification of the CAD using recurrent CNN from CT angiography images. The authors employed 3D-CNN to extract features, and then used recurrent CNN to classify the extracted features. The authors trained this combination on 98 angiography images and tested it on 65 angiography images. The authors achieved an accuracy of 77% for the classification. Han et al. (2020) proposed DL analysis for CAD detection using DL models. In this study, the authors used data from 100 angiography patients for training and 50 for testing the model. The authors proposed an AI system for CT angiography classification. The authors statistically analyzed coronary images in this study. The authors achieved accuracy of 86%, sensitivity of 83%, and specificity of 88%.

All the models mentioned above achieved lower accuracy and utilized fewer image datasets for diagnosing CAD. These models lacked the use of XAI to interpret the networks’ decisions. Current CAD detection methods demand substantial time and computing resources for training before producing acceptable results. Identifying significant patterns in an image requires valuable attributes. The latest models face challenges in overcoming underfitting and overfitting. To address these issues, the ConvMixer was introduced, incorporating MLFR and Morpho operations for automatic CAD diagnosis using patch extraction, similar to the ViT and MLP-mixer. This work can diagnose CAD efficiently, reducing the burden on medical practitioners.

Problem statement

Globally, CAD accounts for a significant portion of the fatalities caused by cardiovascular diseases and is thus the leading cause of morbidity and premature mortality. Most people believe that invasive ICA and other traditional diagnostic procedures are the gold standard. However, they are costly, resource-intensive, and subjective, making it more difficult to make sound therapeutic recommendations. However, current CNNs still struggle with processing overhead, model interpretability, and working with data from disparate sets, despite DL’s impact on medical picture detection. This article proposes a different approach to categorizing CAD. Using ConvMixer, a lightweight neural architecture affected by ViTs and MLP-Mixer, it quickly pulls out spatial and contextual information from coronary CT angiography images. This study utilizes advanced preprocessing techniques, including MLFR and morphological processes, on a large dataset of 5,959 angiographic images to improve edge definition and reduce noise. The ConvMixer model loses less data than other CNN designs because it uses direct patch processing to keep the integrity of the space.

Datasets and methodology

Datasets and pre-processing

This dataset consists of coronary arteries of 500 people. Each picture depicts a mosaic projection view (MPV), including 18 distinct images of a straightened coronary artery arranged vertically. The training, validation, and test sets are divided in to 8:1:1 ratio, with each set comprising 50% normal cases and 50% sick case (Gupta et al., 2020). This dataset includes a total of 5,959 CT angiography images. Out of these, 2,539 are positive and 3,420 are negative images. A random selection of 4,827 images are used for training, 536 for validation, and 596 for testing. MLFR and Morpho techniques were used for pre-processing, which are discussed in detail in the following sections.

Methodology for the classification of the CAD using ConvMixer

This section presents the proposed model for the automated diagnosis of CAD using ConvMixer, based on 5,959 CT angiography images. Figure 1 shows the pipeline for the diagnosis of the CAD using convMixer and XAI applied to CT angiography images. This classification of CAD has the following steps: Collection of data: In this, cardiovascular CT angiography images are collected from publically available platform Mendeley Data (Gupta et al., 2020).

Pre-processing of CT angiography images: In this, two different techniques, MLFR and Morpho operations, were applied to enhance and sharpen the edges of the collected images. Median filtering (MLFR): Random noise can be blocked by median filters, especially when the noise amplitude probability distribution has big tails and regular patterns. The median filtering method is executed by traversing a window over the picture. The filtered picture is produced by positioning the median of the values inside the input window at the middle location of that window in the output image. The median serves as the greatest probability estimate of location for Laplacian noise distribution. The median filter does a good job of predicting the gray-level value in areas that are mostly the same, especially when there is long-tailed noise. Upon crossing an edge, one side predominates the window, resulting in a sudden transition between values in the output. Consequently, the boundary remains distinct. Some problems with these filters are that they can mess up the edges of the picture and add extra noise when the signal-to-noise ratio is low, and they cannot get rid of medium-tailed (Gaussian) noise distributions. In digital image processing, MLFR is particularly popular because, in some cases, it removes noise nevertheless retain edges.

Morphological operations (Morpho): Morphology encompasses a comprehensive array of image processing techniques that manipulate pictures according to their forms. Morphological procedures apply a structuring element to an input picture, resulting in an output image of the same dimensions. A morphological process determines the value of each pixel in the output picture by comparing its corresponding pixel in the input image with its neighboring pixels. The fundamental Morpho operations are dilation and erosion. Dilation increases the pixel count along the edges of objects in a picture, whereas erosion decreases the pixel count at object borders. The number of pixels added or removed from objects in a picture is dependent on the dimensions and configuration of the structuring element used in image processing. In dilation and erosion processes, the associated pixel and its adjacent pixels in the input image dictate the status of each pixel in the output picture through a rule. The rule for manipulating the pixels defines the operation as either dilation or erosion.

Open and close processes are performed on the images based on the dilation and erosion operations. The opening procedure first erodes an image, then dilates the degraded image, employing the same structural element for both processes. A Morpho opening effectively removes small things and thin lines from a picture while maintaining the shape and size of larger objects. The closing procedure expands an image and subsequently contracts the dilated image, employing the identical structural element for both processes. Morpho closure effectively fills small gaps in a picture while maintaining the form and dimensions of larger voids and objects. The example for the pre-processed images using MFLR and Morpho are presented in the Fig. 2.

The MLFR and Morpho approaches in the pre-processing phase provide several benefits compared to traditional procedures, rendering them exceptionally successful for picture augmentation. MLFR efficiently eliminates impulsive noise, such as salt-and-pepper noise, while maintaining essential edges and structures, in contrast to linear filters that sometimes obscure significant features. This guarantees that critical properties are preserved for further processing. Morpho procedures improve the structural integrity of pictures by refining object borders, removing minor undesired artifacts, and filling gaps. The integration of dilation and erosion aids in maintaining essential objects while eliminating extraneous noise, rendering it especially beneficial for activities necessitating accurate shape and edge information. Morpho-based opening and closing processes also improve segmentation by reducing errors, which is important for deep learning (DL) models that depend on clear, defined features. The MLFR and Morpho methods guarantee clear, noise-free, and well-structured pictures, unlike common pre-processing methods like Gaussian filtering, which can cause a lot of blurring, or Fourier-based methods, which may lose spatial information. This finally improves the precision and dependability of classification tasks by supplying high-quality input data for DL models.

Data spliting: After image pre-processing, all images are splitted into training, validation, and test data in the ratio of 8:1:1, respectively.

Figure 1 Proposed coronary artery disease classification diagram.

Figure 2 Example of CAD CT-images with image pre-processing techniques (MFLR and Morpho).

Structure of ConvMixer

It is possible that patch-based representation is more responsible for the impressive performance of visual transformers than the transformer architecture itself. This work proposed a straightforward convolutional architecture for CAD classification, which refer to as ConvMixer. This architecture bears many similarities to the ViT. For example, it works directly on patches, keeps the same representation size and resolution across all layers, doesn’t downscale the representation at lower layers, and can tell the difference between channel-wise mixing and spatial mixing of information. On the other hand, in contrast to the ViT and the MLP-Mixer, this design exclusively uses ordinary convolutions to perform all of these functions. This structure of ConvMixer is presented in the Fig. 3.

Figure 3 Architecture of the ConvMixer (Trockman & Kolter, 2022).

ConvMixer comprises a patch embedding layer succeeded by many iterations of a basic fully-convolutional block, maintaining the spatial configuration of the patch embeddings (Liu et al., 2024). This is represented as:

(1) CVMO=Batch_Norm(σCvcin→ho(Xin,S=p,K_size=p))

where, p is patch size, S is stride, K is kernel, Xin is input, and CVMO is the patch embedding of the convMixer. This block comprises depth-wise convolution (CDW) succeeded by point-wise convolution (CPW). It is most effective with exceptionally large kernel sizes for the CDW. Every convolution is succeeded by an activation function and subsequent Batch Normalization:

(2) CVMl′=Batch_Norm(σCDW(CVMl−1))+CVMl+1

(3) CVMl+1=Batch_Norm(σ(CPW(CVMl′))).

Following several executions of this block, carry out global pooling to obtain a feature vector of size h, that is subsequently input into a softmax. In this work, the convMixer_256_8 model was used for classification with specification listed in the Table 1. Where, eight is the depth of the network, 256 is the number of channels, and it has only 0.8 million parameters. Due to this light weight computational complexity reduces, and less time will take for the training. ConvMixer has total 37,558,600 parameters, out of these 9,973,762 are trainable parameters, 7,637,312 are non-trainable parameters, and 19,947,526 are optimizer parameters. The convMixer_256_8 require 38.05 MB of size to train and test the model. The ConvMixer model specifications are defined and then trained using the parameters listed in Table 1. Based on Trockman & Kolter (2022), parameters were chosen for model training to provide the best performance, stability, and generalization. Since too big a learning rate might lead to divergence and too small a rate can slow down training, the learning rate of 0.003 is used to strike a compromise between stability and quick convergence. We used the Adam optimizer due to its ability to adjust learning rates, which effectively manages sparse gradients and accelerates convergence. A batch size of 128 helped keep the gradient updates stable, and 30 epochs let us fine-tune the network for specific tasks while also making sure the model is well-trained and avoiding overfitting. In order to reduce computational complexity, the sparse categorical cross-entropy loss function was used for classification. We used a kernel size of 5 and a patch size of 2 to maintain computational efficiency while capturing spatial characteristics. We selected eight convMixer layers for depth to minimize overfitting, reduce complexity, and provide adequate feature learning capability. The smoother non-linearity of the GeLU activation function, which improves gradient flow and learning dynamics, made it the best choice over the ReLU. A weight decay of 0.0001 was also used as a regularization method to guarantee improved generalization and avoid overfitting. We used the above mentioned parameters to give best and robust model for the classification. To evaluate its efficacy, the proposed model is tested on a separate set of images, with performance assessed through a confusion matrix. Additionally, XAI techniques—including GCAM, LIME, and OS are applied to the test images to interpret and visualize the model’s decision-making process.

Table 1 Parameters and specification used for the model training.

Parameters or specifications	Value	
Weight decay	0.0001	
Learn rate	0.003	
Batch size	128	
Epochs	30	
Channels or filters	256	
Loss	Sparse categorical cross entropy	
Patch size	2	
Kernel size	5	
Depth	8	
Optimizer	Adam	
Activation	GeLU	

Advantages of convMixer

The main advantages of convMixer compared to MLP-mixer, ViT, and CNNs are as follows: It has simple and isotropic architecture compare to ViT.

The patch dimension will same throughout the processing in the convMixer where as dimension will reduce in the MLP-mixer and ViT.

Point-wise and depth-wise convolutions are performed.

It is an simple CNN structure made up with convolutions, batch normalization, and activations.

Due to its simple architecture, it has less computational complexity.

Experimental results and discussions

In this section, the classification performance of the proposed model is described. In this work, first the convMixer, MLFR+convMixer, and Morpho+convMixer models were trained on 4,827 images using parameter listed in the Table 1 and Fig. 4 shows the training accuracy and loss curve of the proposed model. After training, classification of 596 images are taking place. Accuracy, sensitivity, precision, specificity, JCI, kappa, MC, and F-score have been calculated using the confusion matrices. Figures 5 and 6 show the patch extracted from the proposed model and activations of the convolutional layer of the proposed Morpho+convMixer model respectively.

Figure 4 Model accuracy and loss curves.

Figure 5 Patch embedding of the ConvMixer.

Figure 6 Activation’s of the ConvMixer in convolutinal layer kernels.

Figure 7 gives the confusion matrices of the convMixer, MLFR+convMixer, and Morpho+convMixer models. From Fig. 7, 319 images from positive and 246 images from negative images are truly predicted using convMixer, 309 images from positive and 221 images from negative are correctly detected using MLFR+convMixer, and 337 images from positive and 237 images from the negative are correctly classified using Morpho+convMixer. From these confusion matrices, the performance of the proposed model were evaluated in the Table 2. Table 2 represents the performance of the proposed models. In this, convMixer achieved 94.63%, 95.82%, 93.10%, 94.69%, 90.93%, 80.32%, 89.08% and 95.25% of accuracy, sensitivity, specificity, precision, JCI, kappa, MC and F-score, respectively. MLFR+convMixer achieved 88.92%, 89.56% 88.04%, 91.15%, 87.70%, 64.57%, 77.37% and 90.35% of accuracy, sensitivity, specificity, precision, JCI, kappa, MC, F-score respectively. Morpho+convMixer achieved the highest accuracy of 96.30%, 94.39% of sensitivity, 99.16% of specificity, 99.41% of precision, 93.87% of JCI, 86.78% of kappa 92.58% MC, and 96.83% of F-score. From these, it can be concluded that this proposed model convMixer with morphological operations model performed better than other two models listed in the Table 2. Figure 8 presents a visual representation of the performance evaluation of the proposed ConvMixer models. Figure 9 shows the receiver operating characteristic (ROC) of the proposed model.

Figure 7 Confusion matrices for the CAD classification using proposed models.

Table 2 Performance evaluation of the proposed ConvMixer models.

Method	Accuracy	Sensitivity	Specificity	Precision	JCI	Kappa	MC	F-score	
ConvMixer	94.63	95.82	93.10	94.69	90.93	80.32	89.08	95.25	
MFLR+ConvMixer	88.92	89.56	88.04	91.15	87.70	64.57	77.37	90.35	
Morpho+ConvMixer	96.30	94.39	99.16	99.41	93.87	86.78	92.58	96.83	

Figure 8 Graphical representation of performance of different ConvMixer models.

Figure 9 ROCs of of the proposed models.

XAI using GCAM, LIME and OS

Figure 10 shows that XAI is applied to both positive and negative images in order to understand the decision made by the proposed model. In this study, GCAM, LIME, and OS were applied to the images. The network extracts more discriminatory features from the image that is represented in dark red, fewer features are extracted from the image that is shown in light red, and no features are extracted from blue, allowing it to make decisions based on the model’s extracted features.

Figure 10 Example of XAI (GCAM, LIME, and OS) decision understanding by proposed model.

Limitations of the proposed work or research

The suggested model provides a simple architecture for CAD categorization. Although it demonstrates competitive efficacy in specific tasks, it possesses several drawbacks relative to more sophisticated systems such as CNNs or Transformers. It is deficient in multi-scale feature extraction and hierarchical representation capabilities compared to typical CNNs, rendering it less effective for intricate applications. In contrast to ViTs, it lacks attention mechanisms, hence constraining its capacity to capture long-range relationships. It may also exhibit suboptimal performance on limited datasets and need extended training durations or more data augmentation to get competitive outcomes. Moreover, its diminished inductive bias and restricted utilization in research render it less adaptable than recognized designs such as CNNs or Transformers. Notwithstanding its efficiency, these constraints constrain its application to more complex or large-scale picture categorization tasks (Demirbaş, Üzen & Fırat, 2024).

These limitations can be overcome in the future by utilizing the attention mechanism in the simplest way, using a large amount of datasets, using real-time data images, using advanced optimization techniques, and also using multi-modal datasets. The model’s attention mechanisms will effectively address the inductive bias issue.

Comparison of the proposed model with previous state-of-the-art models and other DL-models

Table 3 presents the comparison of existing state-of-the-art (SOTA) methods with proposed model. In Jin et al. (2022), the authors attained a maximum accuracy of 97.00% employing a CNN+GBDT methodology with a dataset of 890 samples. Nevertheless, the authors attained a sensitivity of 84.10%, suggesting possible constraints in identifying positive cases. In a similar vein, Sayadi et al. (2022) documented an accuracy of 95.45% with an SVM model applied to the Z-Alizadaly dataset, whereas Abdar et al. (2019) attained an accuracy of 93.08% utilizing a N2Genetic-NVSVM model. Alternative SVM-based approaches, including Saeedbakhsh et al. (2023) and Garavand et al. (2022), attained accuracies of 89.73% and 88.00%, respectively, although they did not assess specificity or sensitivity. In Zreik et al. (2018a), a 3D CNN+RCNN model was employed, achieving a modest accuracy of 77.00%. However, their subsequent research in Zreik et al. (2019) enhanced this to 87.00% by utilizing a conventional CNN model. In Han et al. (2020), the CNN achieved an accuracy of 86.00%, with a specificity of 88.00% and a sensitivity of 83.00%. Conversely, in Muscogiuri et al. (2024), the accuracy was 91.50%, with a specificity of 95.30% and a sensitivity of 79.70%. The suggested model surpasses the majority of current strategies, with an accuracy of 96.30%, a specificity of 99.16%, and a sensitivity of 94.39%, so illustrating its robust capacity to differentiate among various classes. It was trained on a considerably larger dataset of 5,959 samples, enhancing its robustness and generalizability relative to models developed on smaller datasets. The findings demonstrate that the suggested method offers enhanced accuracy while achieving an improved equilibrium between specificity and sensitivity, rendering it a more effective and dependable strategy for classification tasks.

Table 3 Comparison of existing SOTA with proposed model.

References	Method	Dataset	Accuracy	Specificity	Sensitivity	
Abdar et al. (2019)	N2Genetic-NVSVM	–	93.08	–	–	
Saeedbakhsh et al. (2023)	SVM	11,495	89.73	–	–	
Sayadi et al. (2022)	SVM	Z-Alizadaly	95.45	–	–	
Garavand et al. (2022)	SVM	328	88.00	–	–	
Jin et al. (2022)	CNN+GBDT	890	97.00	95.70	84.10	
Zreik et al. (2019)	CNN	379	87.00	–	–	
Zreik et al. (2018a)	3D CNN+RCNN	163	77.00	–	–	
Han et al. (2020)	CNN	150	86.00	88.00	83.00	
Muscogiuri et al. (2024)	CNN	–	91.50	95.30	79.70	
Proposed	Morpho+ConvMixer	5,959	96.30	99.16	94.39	
Note:

GBDT, Gradient boosting and decision tree.

Table 4 represents the comparison of DL techniques with the proposed model. ResNet50 achieved an accuracy of 94%, with a precision of 84% and sensitivity of 85% across the models. Conversely, AlexNet has the lowest performance, achieving an accuracy of 84%, a precision of 69%, and a sensitivity of 66%, demonstrating its constrained efficacy. Xception demonstrates an enhanced precision of 88%, yet a slightly reduced sensitivity of 81%, culminating in an F-score of 83%. Recurrent models like LSTM and BiLSTM work better than others. For example, BiLSTM has a sensitivity of 90%, which is higher than LSTM’s sensitivity of 85% and leads to better overall effectiveness. The MLP mixer and ViT models achieved an 94.00% and 95.00% of accuracy, respectively. The proposed ConvMixer model, which combines morphological processing with ConvMixer, surpasses all existing models. It attains a maximum accuracy of 96.30%, an outstanding precision of 99.41%, and a remarkable F-score of 96.83%, establishing it as the most successful method. The markedly superior precision and F-score demonstrate that the suggested method improves classification performance, giving it a more dependable and effective alternative to traditional DL-techniques. This is attributed to its straightforward architecture and specifications. The graphical representation of this comparison is shown in the Fig. 11.

Table 4 Comparison proposed model with other DL-techniques.

Method	Accuracy	Precision	Sensitivity	F-score	
ResNet50	94.00	84.00	85.00	84.00	
AlexNet	84.00	69.00	66.00	66.00	
Xception	87.00	88.00	81.00	83.00	
GoogleNet	87.22	77.00	73.00	75.00	
VGG19	84.00	78.00	68.00	70.00	
LSTM	92.00	81.00	85.00	82.00	
BiLSTM	93.00	83.00	90.00	85.00	
MLP Mixer	94.00	86.00	82.00	84.00	
ViT	95.00	94.00	94.00	94.00	
Proposed (Morpho+ConvMixer)	96.30	99.41	94.49	96.83	

Figure 11 Performance of the ConvMixer and other DL-methods.

The suggested model fundamentally consists of an MLP-Mixer augmented with convolutions (Tolstikhin et al., 2021). It operates directly on embedded patches, guaranteeing consistent resolution and dimensions throughout the layers. In addition, depth-wise separable convolution can tell the difference between channel-wise and spatial information integration, just like MLP-Mixer, and it has similar skip connections. The suggested framework is a complete CNN. All ConvMixer actions may be performed just using activations, batch normalization, and convolutions. Consequently, it is fundamentally a CNN with certain architectural hyper-parameters. Whereas ViT performs well for large datasets and high-resolution images, CNN performs well for small and medium datasets, and it also provides good real-time interference. The proposed model utilizes the advantages of CNNs, MLP mixers, and ViT models to give the best performance in CAD detection.

In the future, advanced attention mechanisms with ViTs, such as locally shifted attention tokenization, will be applied for image classification and feature extraction. Additionally, conformer or Multi-modal Adaptive Model-based Biomarker Analysis (MAMBA) techniques will be utilized to enhance diagnostic efficacy in classification tasks.

Conclusion

The coronary vascular imaging reveals that the artery is a slender, tubular structure with relatively low contrast and artifacts. This complicates the accurate classification of the samples. This article suggested a DL method that uses morphological operations and ConvMixer to classify the coronary blood vessels in CT angiography images. This study introduces ConvMixer, which uses a median filter and morphological methods to pre-process CT angiography images into groups based on coronary artery disease. The proposed model utilizes the advantages of the MLP mixer, CNNs, and ViT. The Marpho+ConvMixer model keeps the resolution the same throughout the process, like CNNs. It also uses depth-wise separable convolutions, like the MLP-mixer, and patches for robust feature extraction that are lighter than the ViT. The proposed model implements all these processes through convolutions. Due to this, the proposed model gives robust feature extraction with less computational cost as compared to other models. We used 5,959 CT angiography images for categorization purposes.For the combination of morphological operations and ConvMixer, an accuracy of 96.30%, sensitivity of 94.39%, and specificity of 99.16% was achieved; for ConvMixer alone, 94.63% accuracy, 95.82% sensitivity, and 93.10% specificity were achieved; and for the combination of median filter and ConvMixer, 88.92% accuracy, 89.56% sensitivity, and 93.10% specificity were achieved. The results show that it is possible to automatically and non-invasively find patients who need invasive coronary angiography. They also show that future coronary artery procedures are possible. This may potentially reduce the number of individuals undergoing invasive coronary angiography. Finally, we conducted post-image analysis using DL heat maps to understand the decisions made by the proposed model. The proposed integrated DL intelligent system improves diagnosis accuracy, reduces the need for medical staff, reduces manual work in diagnosis, and offers additional methods to monitor coronary angioplasty-related medical diagnostic systems. This model also improves the generalizability, accuracy, and interpretability in order to detect the CAD automatically.

Supplemental Information

Supplemental Information 1 Sample dataset.

Additional Information and Declarations

Competing Interests

The authors declare that they have no competing interests.

Author Contributions

C. Rajeev conceived and designed the experiments, performed the experiments, analyzed the data, performed the computation work, prepared figures and/or tables, and approved the final draft.

Karthika Natarajan conceived and designed the experiments, performed the experiments, analyzed the data, performed the computation work, authored or reviewed drafts of the article, and approved the final draft.

Data Availability

The following information was supplied regarding data availability:

The dataset used in this study is available at Mendeley: Demirer, Mutlu; Gupta, Vikash; Bigelow, Matthew; Erdal, Barbaros; Prevedello, Luciano; White, Richard (2019), “Image dataset for a CNN algorithm development to detect coronary atherosclerosis in coronary CT angiography”, Mendeley Data, V1, doi: 10.17632/fk6rys63h9.1.

Additional details about the dataset is available in the published article: https://doi.org/10.1007/s10278-019-00267-3.

A sample dataset is available in the Supplemental File for reference.

The raw data processing code is available at GitHub and Zenodo:

- https://github.com/Rajeev24-cmd/CAD-classification-using-convmixer.

- Rajeev24-cmd. (2025). Rajeev24-cmd/CAD-classification-using-convmixer: CAD-classification-using-convmixer (CON). Zenodo. https://doi.org/10.5281/zenodo.14850693.

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
