# Peer review of "Coronary artery disease classification using ConvMixer based classifier from CT angiography images"

_PeerJ Computer Science, doi:10.7717/peerj-cs.2771_

## Round 0.1 · original submission · Major Revisions

Dear authors,
You are advised to critically respond to all comments point by point when preparing an updated version of the manuscript and while preparing for the rebuttal letter. Please address all comments/suggestions provided by reviewers, considering that these should be added to the new version of the manuscript.

Kind regards,
PCoelho

Reviewer 1 ·

Basic reporting

The workflow is described clearly.
The literature review is comprehensive.

Experimental design

The relevant metrics are reported.

Validity of the findings

I suggest adding a comparative table detailing the existing techniques, datasets used compared to their dataset and the reported metrics compared to the current work
I suggest adding the limitations of the performed study.
It's not clear if the models were evaluated on an external test set. If yes its important to emphasize that
I suggest emphasizing the important open challenges

Additional comments

The literature review is comprehensive and thorough

Reviewer 2 ·

Basic reporting

All comments are included in detail in the fourth section.

Experimental design

All comments are included in detail in the fourth section.

Validity of the findings

All comments are included in detail in the fourth section.

Additional comments

Review Report for PeerJ Computer Science
(Coronary artery disease classification using ConvMixer based classifier from CT angiography images)

1. Within the scope of the study, various classification operations were performed with the proposed deep-based model using computed tomography angiography images.

2. In the introduction section, coronary artery disease, the importance of the subject and the main contributions of the study were mentioned in sufficient detail.

3. In the literature review section, both machine learning and deep learning-based approaches in the literature related to coronary artery were explained at an appropriate level in terms of their results and contributions to the literature.

4. The dataset used in the study is sufficient in terms of both quantity and type. However, the training, validation and test distributions are stated as 80%, 10% and 10%, respectively. The results obtained in the classification problems are very dependent on the amount of the test dataset. For this reason, it should be explained in detail how the amount and percentage of the test dataset are determined and why cross-validation, which is frequently used in the literature, should be preferred to reduce dataset dependency.

5. Instead of using the raw dataset, passing it through the Morphological Operations and Median Filtering steps increased the depth of the study. However, although there are many different preprocessing methods used in the literature, why these are preferred should be explained more clearly. In addition, it is recommended to compare the results before and after preprocessing in detail in order to observe the effect of preprocessing on the result more clearly.

6. It should be explained how the parameter values/types such as learning rate, epoch and optimizer given in Table-1 are determined and whether different trials are performed.

7. When the proposed model and its steps and the details specified are examined, it is observed that it has a certain level of originality.

8. When the evaluation metric types obtained for the classification processes performed for coronary artery disease are examined, it is observed that most of the metrics required for the correct analysis of the results are obtained. However, it is definitely recommended to obtain ROC (Receiver operating characteristic) curve and AUC (the area under the ROC curve) scores.

9. Table-3 and Table-4 provide a comparison of the proposed model with some other deep learning-based models in the literature. In this section, a comparison with a few more up-to-date deep learning models that can be used especially in biomedical image classification will further increase the quality of the study.

In conclusion, this study proposes a deep learning based model with significant potential for the classification of coronary artery disease with artificial intelligence. However, in order to make a full contribution to the literature, all the above mentioned sections should be addressed in detail.

Reviewer 3 ·

Basic reporting

All basic reporting items were cleanly stated

Experimental design

1. The main problem is that it appears that the paper does not start with a hypothesis to test, or research question to answer but rather focused solely on applying a machine learning method to the data. Author can write research question to answer
2. Authors need write research objective
3. Authors need to explain data pre-processing clearly (i.e data cleansing, noise removal, object detection)

Validity of the findings

1. Some of the existing similar studies have reported higher performance compared to this study, therefore it is necessary to include a discussion on the difference.
2. Authors need to write limitation of this research

---

## Round 0.2 · Minor Revisions

Dear authors,
Thanks a lot for your efforts to improve the manuscript.
Nevertheless, some concerns are still remaining that need to be addressed.
Like before, you are advised to critically respond to the remaining comments point by point when preparing a new version of the manuscript and while preparing for the rebuttal letter.

Kind regards,
PCoelho

Reviewer 1 ·

Basic reporting

all comments were addressed

Experimental design

all comments were addressed

Validity of the findings

all comments were addressed

Additional comments

all comments were addressed

Reviewer 2 ·

Basic reporting

All comments are included in detail in the fourth section.

Experimental design

All comments are included in detail in the fourth section.

Validity of the findings

All comments are included in detail in the fourth section.

Additional comments

Review Report for PeerJ Computer Science
(Coronary artery disease classification using ConvMixer based classifier from CT angiography images)

Thank you for the revision. The responses to the comments and the changes in the paper are sufficient. I recommend that the paper be accepted due to the potential contribution of the study to the literature and its originality at a certain level. Best regards.

Reviewer 3 ·

Basic reporting

English, literature, picture clear

Experimental design

Authors need write research questions clearly
Authors need to explain data preprocessing tasks and result

Validity of the findings

Authors need to write limitatiion of research
Some similar research already similat task, authors need to write novelty of research

---

## Round 0.3 · accepted · Accept

Dear authors, we are pleased to verify that you meet the reviewer's valuable feedback to improve your research.

Thank you for considering PeerJ Computer Science and submitting your work.

Kind regards
PCoelho